# Why do acute healthcare staff engage in unprofessional behaviours towards each other and how can these behaviours be reduced? A realist review protocol

Jill Maben [1], Justin Avery Aunger [1], Ruth Abrams,[1] Mark Pearson [2], Judy M Wright [3], Johanna Westbrook,[4] Russell Mannion,[5] Aled Jones [6]

¹School of Health Sciences, Faculty of Health and Medical Sciences, University of Surrey, Guildford, UK
²Wolfson Palliative Care Research Centre, Hull York Medical School, Hull, UK
³Faculty of Medicine and Health, University of Leeds, Leeds, UK
⁴Australian Institute of Health Innovation, Macquarie University, Sydney, New South Wales, Australia
⁵Health Services Management Centre, University of Birmingham, Birmingham, UK
⁶School of Healthcare Sciences, Cardiff University, Cardiff, UK

**Correspondence to**
Professor Jill Maben;
j.maben@surrey.ac.uk

## ABSTRACT

**Introduction** Unprofessional behaviours encompass many behaviours including bullying, harassment and microaggressions. These behaviours between healthcare staff are problematic; they affect people's ability to work, to feel psychologically safe at work and speak up and to deliver safe care to patients. Almost a fifth of UK National Health Service staff experience unprofessional behaviours in the workplace, with higher incidence in acute care settings and for staff from minority backgrounds. Existing analyses have investigated the effectiveness of strategies to reduce these behaviours. We seek to go beyond these, to understand the range and causes of such behaviours, their negative effects and how mitigation strategies may work, in which contexts and for whom.

**Methods and analysis** This study uses a realist review methodology with stakeholder input comprising a number of iterative steps: (1) formulating initial programme theories drawing on informal literature searches and literature already known to the study team, (2) performing systematic and purposive searches for grey and peer-reviewed literature on Embase, CINAHL and MEDLINE databases as well as Google and Google Scholar, (3) selecting appropriate documents while considering rigour and relevance, (4) extracting data, (5) and synthesising and (6) refining the programme theories by testing the theories against the newly identified literature.

**Ethics and dissemination** Ethical review is not required as this study is a secondary research. An impact strategy has been developed which includes working closely with key stakeholders throughout the project. Step 7 of our project will develop pragmatic resources for managers and professionals, tailoring contextually-sensitive strategies to reduce unprofessional behaviours, identifying what works for which groups. We will be guided by the 'Evidence Integration Triangle' to implement the best strategies to reduce unprofessional behaviours in given contexts. Dissemination will occur through presentation at conferences, innovative methods (cartoons, videos, animations and/or interactive performances) and peer-reviewed journals.

**PROSPERO registration number** CRD42021255490.

## Strengths and limitations of this study

⇒ This review is the first to identify the causes of a wide range of unprofessional behaviours (not only bullying and harassment) among staff for a range of staff groups and contexts in practice.

⇒ The review uses a context-sensitive realist methodology that accounts for complexity, to understand how and why strategies to reduce staff-to-staff unprofessional behaviours in acute care settings work and for whom.

⇒ Reviewing both the causes of unprofessional behaviours and the strategies to reduce them will enable us to gain a better understanding of how strategies interact with and target the underlying mechanisms that lead to unprofessional behaviours.

⇒ This review includes stakeholder engagement throughout, including refinement of programme theory and co-creation of actionable, context-specific outputs.

⇒ Limitations of the review include a primary focus on acute healthcare settings, the inclusion of sources published only in English and focus on health systems with similarities to a UK healthcare context.

## INTRODUCTION

Unprofessional behaviours can negatively impact staff well-being, patient safety and have organisational cost implications.[1 2] They are a pervasive issue in workplaces throughout global healthcare systems.[2–4] The term unprofessional behaviours is a generic one which includes a range of more specific behaviours such as uncivil, transgressive or disruptive behaviours, physical and verbal aggression[5] and bullying.[6] Unprofessional behaviours have been defined as 'a wide spectrum that includes conduct that more subtly interferes with team functioning, such as poor or ambiguous communication, passive aggression, lack of responsiveness, public criticism of colleagues and humour at others' expense'.[7] As such, unprofessional behaviours can be casual and generalised, or highly targeted with the intention to cause harm. Unprofessional behaviours are increasingly being

recognised as unacceptable in a modern, inclusive health setting, as they detrimentally affect the work and psychological well-being of others.

## Prevalence of unprofessional behaviours in the UK healthcare system

Workplace bullying affects staff in all roles and at all levels throughout the UK healthcare system. The 2020 UK National Health Service (NHS) staff survey found that 18.7% of all staff reported experiencing bullying or harassment at work from their colleagues,[8] and 12.4% of staff reported experiencing these behaviours from managers.[8] Staff in ambulance service trusts and certain minority groups experience above average rates of bullying and harassment,[9] including women, individuals with a disability and members of minority communities, through microaggressions[6 10] and harassment or discrimination.[11–13] For example, the 2020 staff survey showed that minority ethnic respondents reported 5.6% higher rates of bullying and harassment from other staff than did white respondents.[8]

## Impacts of unprofessional behaviours

Unprofessional behaviours can have negative impacts on staff psychological well-being in the workplace, on patient safety and quality of care and for wider systems and culture.[14–17] For example, bullying of individual staff members in the healthcare workplace can lead to psychological issues such as depression, burnout and distress.[18] Bullying can also lead to physical problems such as sleep disturbance, headache and gastrointestinal upset, resulting in staff taking sick leave.[2] In extreme cases, bullying and harassment can lead to suicidal ideation.[19]

Unprofessional behaviours can result in staff feeling unsafe at work and an absence of psychological safety may encourage a climate of unprofessional behaviours and inhibit speaking up.[1] Psychological safety at work is absent in teams where incivility, transgressive behaviours and bullying and harassment are experienced.[20] This can result in reduced staff psychological well-being at work and reduced job satisfaction, engagement and motivation.[21]

The impact of unprofessional behaviours on speaking up and patient safety is well documented.[1 22 23] Where a culture of psychological safety is encouraged, employees are more likely to speak up about errors in patient care.[23] Psychologically safe teams share significant information and perform better together, improving patient safety.[24–26] For example, a recent simulation study conducted with nursing students in the USA found that where there was bullying, the work environment became chaotic, overwhelming and increased the prevalence of unsafe practices.[27] Another study performed in Israel found that team-targeted rudeness alone explained 12% of the variance in team care performance with significant negative clinical outcomes.[25]

Unprofessional behaviours are associated with higher staff turnover and other significant financial costs such as litigation, lower staff productivity and sickness leave.[28–31] Kline and Lewis' conservative estimate of the financial cost of bullying and harassment to the NHS (due to sickness absence, employee turnover, reduced productivity, compensation and litigation costs) suggests damages from bullying alone was at least £2.28 billion per annum, or 1.52% of the NHS' budget for 2019/2020.[14]

## Strategies to reduce unprofessional behaviours

Unprofessional behaviours, such as bullying, constitute a complex problem that require a broad-ranging, strategic approach that incorporates multiple levels of analysis to successfully reduce them.[7] One systematic review of workplace incivility in nursing, for example, identified many potential strategies[32]; these included education about workplace incivility and its impacts, assertiveness training, rehearsal of responding to incivility and team building exercises. There are also policy-level initiatives, such as legislation against harassment that is currently enshrined in UK and European law.[33] A recent Cochrane review[33] identifies that strategies can operate on society/policy, organisation, job/task and individual/job interface levels. Such strategies can also be preventative, ameliorative or reactive, in nature.[33]

To date, much of the policy focus with regard to addressing unprofessional behaviours in healthcare settings has been on identifying 'bad apples' (individual instigators), with a more recent focus on the role of the organisational environments and cultures in which they work ('bad barrels').[34] There is also a role for healthcare professions regulators to regulate their own organisations ('bad cellars'), as well as considering the shifting wider political and policy context ('bad orchards').[34] Capturing these different levels of analysis and interventions is a complex task, requiring a methodology capable of understanding how strategies interact to produce particular outcomes in different contexts and for whom. A realist approach is one such methodology.

## Existing syntheses and our contribution

Our realist review will complement and extend other recent reviews.[35 36] These include, for example, a systematic review which identifies the predictors and triggers of staff incivility within healthcare teams, including high workload, communication issues, lack of support, poor leadership and being more junior in a team.[35] Similarly, a realist-informed review of bullying and harassment concluded that the involvement of management and wider organisational commitment were key to intervention success.[36] While this review focused solely on bullying and harassment,[36] others have mapped the terrain without providing sufficient granular detail to differentiate between professional groups and contexts, or have included unprofessional behaviours towards patients.[37] They have primarily investigated the effectiveness of strategies for reducing unprofessional behaviours but do not always explain why they do or do not work.

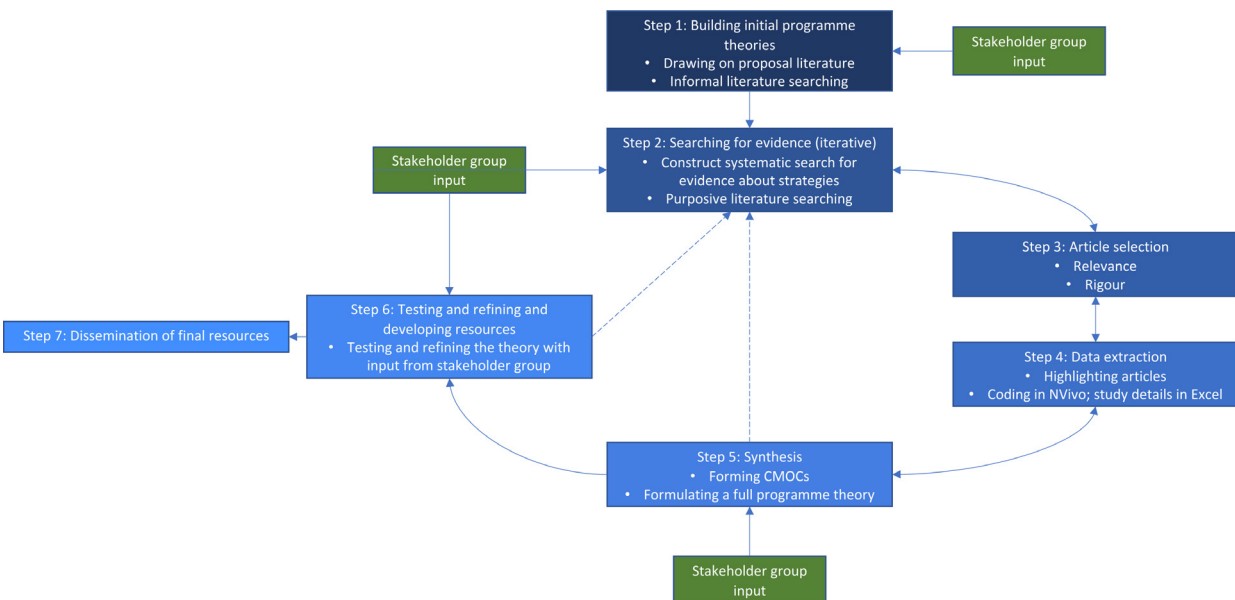

**Figure 1** Review flow diagram. Dotted line reflects elements that may potentially happen. CMOCs, context, mechanism and outcome configurations.

Our realist review differs from these by focusing on unprofessional behaviours among all staff to produce more detailed knowledge regarding the causes of such behaviours as well as improve context-specific understanding of how, why and in what circumstances unprofessional behaviours between staff in acute healthcare settings occur, and how they can best be understood, mitigated, managed and prevented. Lastly, we will look beyond single professions to consider multiple professional groups and associated team-based and organisational interprofessional dynamics, which may impact on how unprofessional behaviours emerge or are addressed.

## METHODS
### Review aim, objectives and research questions
#### Aim
This realist review aims to improve context-specific understanding of how, why and in what circumstances unprofessional behaviours between staff in acute healthcare settings occur, and how strategies can be implemented to mitigate, manage and prevent them.

#### Objectives
This review seeks to:
1. Conceptualise and refine terminology: by mapping behaviours defined as unprofessional to understand differences and similarities between terms referring to unprofessional behaviours (eg, incivility, bullying, microaggressions) and how these terms are used by different professional groups in acute healthcare settings.
2. Develop and refine context, mechanism and outcome configurations (CMOCs): to understand the causes and contexts of unprofessional behaviours; the mechanisms which trigger different behaviours; and the outcomes on staff, patients and wider system of healthcare.

3. Identify strategies designed to mitigate, manage and prevent unprofessional behaviours and explore how, why and in what circumstances these work and whom they benefit.
4. Produce recommendations and comprehensive resources that support the tailoring, implementation, monitoring and evaluation of contextually-sensitive strategies to tackle unprofessional behaviours and their impacts.

#### Research questions
► How are unprofessional behaviours defined, developed and experienced by staff in acute healthcare settings?
► How do current strategies to reduce unprofessional behaviours in acute care address these behaviours or not?
► What are the mechanisms acting at individual, group, professional and organisational levels that underpin strategies aimed at reducing unprofessional behaviours?
► What are the outcomes of unprofessional behaviours on staff (well-being, psychological safety), organisations (recruitment, turnover) and patients (eg, patient safety and care quality)?
► What are the contexts which determine whether the different mechanisms produce their intended outcomes?
► What changes are needed to existing and/or future strategies to make them more effective?

#### Study design
This study uses a realist review methodology. Realist reviews are theory-driven and synthesise literature about complex social interventions.[38] They focus on understanding the mechanisms by which strategies work (or

**Table 1** Inclusion criteria

| Category | Criterion |
|---|---|
| Study design | Any (including non-empirical papers/reports). |
| Study setting | Acute healthcare settings—acute, critical, emergency (and potentially wider, see relevance criteria below). |
| Types of unprofessional behaviour | All as exhibited and experienced by healthcare staff (not patients nor patient to staff). |
| Types of participants | All employed staff groups including students on placements. |
| Types of interventions/strategies | Individual, team, organisational and policy level interventions. Cyber-bullying and other forms of online unprofessional behaviour included if it is staff-to-staff only. |
| Causes of unprofessional behaviours | All. |
| Outcomes | Included but not limited to all sources that focused on one or more of the following aspects: staff well-being (stress, burnout, resilience) staff turnover, absenteeism, malpractice claims, patient reports, magnet hospital/recruitment, patient safety (avoidable harm, errors, speaking up rates, safety incidents, improved listening/response), cost. |
| Language | Only sources in English will be included. |

Searches will not be limited (eg, by date or publication) but records tagged as 'children', 'animals' or 'elder abuse' will be removed, and we will seek to exclude papers describing staff to patient or patient to staff unprofessional behaviours.

not) and seek to understand contextual influences on whether, why, how and for whom these might work.[39] In a realist understanding, contexts are either (1) observable features (space, place, people, things) that trigger or block an intervention with activation on a continuum, rather than a binary trigger (on/off switch)[40] or (2) relational and dynamic features that shaped the mechanisms through which an intervention works.[41] Likewise, mechanisms are interpreted as changes to participant reasoning in response to resources (eg, education about unprofessional behaviours) introduced by an intervention.[40]

The realist approach to data collection and analysis is driven by retroduction, a form of logical inference, which starts with the empirical, and explains outcomes and events by identifying the underlying mechanisms which are capable of producing them.[42] It is therefore essential to consider not only the specific unprofessional behaviours within the healthcare workforce, but any differences and similarities between staff groups as well (eg, by specialty, professional group, setting or seniority). By illuminating these contexts and working practices, we will also be able to determine how they might influence the presence or minimisation of unprofessional behaviours between healthcare staff working in acute settings.

This review will follow the Realist and Meta-Review Evidence Synthesis: Evolving Standards guidelines on quality and reporting.[43] An overview of the review process is shown in figure 1, with the steps we will take to achieve our objectives explained in the following sections.

### Public, patient and stakeholder involvement
We have worked closely with a stakeholder group in formulation of the proposal and have built in engagement as a key component to inform theory development. These stakeholders come from a range of relevant backgrounds, including patients and the public, heads of professional

standards bodies, members of regulatory bodies and unions in the UK, influential theorists in the field, healthcare professionals with lived experience of unprofessional behaviours and minority ethnic perspectives. This protocol incorporates four time points where stakeholder feedback will be incorporated (figure 1). This includes sense-checking our initial programme theories, aiding in identifying the most relevant evidence, ensuring our CMOCs reflect lived experience and refining the final programme theories. In line with the ACTIVE (**A**uthors and **C**onsumers **T**ogether **I**mpacting on e**V**idenc**E**) framework for stakeholder involvement in systematic reviews, our methodology is analogous to a continuous, multiple-time closed event approach with stakeholders able to influence the results of the review.[44]

Stakeholder group feedback will be incorporated using the following processes to ensure rigour and relevance: (1) we will record what and how aspects of the developing theory are presented to stakeholders for refinement; (2) we will record where any alterations are suggested; (3) where suggestions are not aligned with what has been identified in the literature, we will perform purposive searching to sense-check recommendations; (4) any remaining discrepancies will be discussed within the team to determine majority consensus and (5) re-presentation of changes made in response to feedback to individual stakeholders or the whole group to further sense-check any changes.[45] We will take detailed notes of the meetings with the aid of a project administrator and will follow-up with stakeholders individually regarding specific points raised.

### Step 1: identifying existing theories
To develop initial programme theories, we will iteratively: (a) Draw on preliminary discussions within the project team, with the healthcare workforce, patients and the

public; (b) Consult with our multidisciplinary stakeholder group as outlined above; (c) Examine healthcare literature already known to the research team (namely papers, reviews and reports identified in our initial scoping review which informed the funding proposal development), informal searching of key organisational websites (eg, King's Fund, BMA, NHS England) and use of the 26 behaviours reported by Westbrook *et al* for comparison against behaviours identified in the literature.[2]

This first informal screening of the literature will sensitise the team to the breadth and depth of published and unpublished literature on unprofessional behaviours within healthcare. By investigating the theoretical underpinnings of programmes, we shall map the conceptual and theoretical landscape of unprofessional behaviour causes, outcomes and how any identified strategies and interventions are theorised to work in acute healthcare settings. Building these initial programme theories will require iterative discussions within the project team and with our stakeholders to make sense of and synthesise the different assumptions. Once the programme theories have been developed by the project team, they will be presented to the stakeholder group to integrate their feedback and to ensure practical applicability (figure 1).

### Step 2: searching for evidence
Realist reviews typically draw on a combination of systematic searches, more informal, purposive searches, citation tracking and reference scanning.[39] It is also common to perform more searches later in the process in an evolutionary manner to further explain certain areas which are found to be underexplored in the programme theory. Additionally, in realist reviews, what is considered evidence often differs from other reviews. For example, non-empirical studies and commentaries can be included if they illuminate the topic being studied or can otherwise contribute to the programme theory. In realist reviews, the purpose is not to necessarily to identify every item of literature that could relate to a phenomenon, but, rather, to develop and refine a comprehensive theory to explain how the intervention works, why, for whom and in what circumstances.

### Search strategy
We will take a systematic approach and conduct searches to identify literature with which to test and refine our initial theories. In this formal, primary search we will: (a) Identify studies addressing strategies to reduce unprofessional behaviours in acute healthcare settings with all healthcare staff. Search academic databases including MEDLINE, Embase and CINAHL and sources of trade, policy and grey literature including Agency for Healthcare Research and Quality's Patient Safety Network and Google Scholar. Examples of trade journals we will search include AACN Bold Voices, Nursing Times, ED Management and Nursing Standard. Search strategies will comprise search terms, synonyms and index terms for: Acute care AND Healthcare staff AND Unprofessional

behaviours. Searches from existing similar reviews such as Iling *et al* will be consulted to aid identifying relevant search terms.[36] See online supplemental file 1 for an indicative example search strategy, which is not complete, as searches are likely to evolve as is typical in a realist synthesis.[39]

(b) Conduct a grey literature search for professional codes of conduct produced by the Royal Colleges and the literature on cases brought to the Nursing and Midwifery Council, Healthcare and Professions Council and General Medical Council for unprofessional behaviour; using databases such as Health Management Information Consortium database and websites including NHS Employers, NHS Health Education England and Google (limited to screening the first 200 results).

Search strategies will be peer-reviewed using the PRESS (Peer Review of Electronic Search Strategies) method.[46] All search results will be saved in reference management software. A detailed, shared spreadsheet will record all searches conducted, to ensure transparency when reporting the search activities.

### Additional searching
If we find that we require more data to develop, confirm, refute or refine programme theory development, we will conduct additional purposive searches using CLUSTER search methods[47] to further develop any areas of the theory which require greater evidence to fully understand. These and earlier searches will follow the iterative realist search methods described by Booth *et al.*[48]

### Step 3: article selection
#### Inclusion and exclusion criteria
We will include both peer-reviewed and grey literature which help understand how and why strategies to reduce unprofessional behaviours in acute care settings work and whom they benefit. The following inclusion criteria will be used (table 1).

### Screening, relevancy and rigour
Screening of search results will be primarily undertaken by JAA in collaboration with JM, MP or RA who will screen a subset of records retrieved. A 10% random subsample of the citations retrieved from searching will be reviewed independently for quality control (by a second reviewer, either JM, MP or RA) at title and abstract, full text and relevancy stages. Any disagreements will be resolved by discussion between JA, the second reviewer and JM. If disagreements remain, then a third member of the team (AJ, RM or JMW) will review and any disagreements will be resolved through further review/discussion, with final decision resting with JM. The remaining 90% of decisions at these stages will be made by JAA. Screening of titles and abstracts will be performed using Rayyan.ai software (http://www.rayyan.ai/) and full texts will be screened using Mendeley (Mendeley).[49]

Beyond the inclusion and exclusion criteria above, decisions regarding the inclusion or exclusion of sources

will be based on a combination of relevance (based on both the major/minor criteria below and the ability to inform programme theories, that is, conceptual richness and depth of sources) and rigour (whether the methods used to generate the relevant data are credible and trustworthy). Assessment of rigour will focus on the extent to which sources provide a detailed description of their methods and how generalisable and trustworthy their findings are based on those methods.[39]

**Box 1    Strategies foundational to the realist analysis**

**Strategies for synthesising evidence**
⇒ Compare and contrast sources of evidence—eg, where evidence about interventions or its mechanisms in one source allows insights into evidence about outcomes in another paper.
⇒ Reconciling of sources of evidence—where results differ in apparently similar circumstances, further investigation is appropriate in order to find explanations for why these different results occurred.
⇒ Adjudication of sources of evidence—this will enable us to make the synthesis more manageable. This includes strategies outlined in 'article selection' section above, ie, dividing papers that make 'major' or 'minor' contributions to our research questions (51).
⇒ Consolidation of sources of evidence—where outcomes differ in particular contexts, an explanation can be constructed of how and why these outcomes occur differently.

Our formal criteria for classifying the potential relevancy of sources are, by ascending order of relevancy:

Major contribution:
► Sources which contribute to the study aims and are conducted in an NHS context in acute care; or,
► Sources which contribute to the study aims and are conducted in an NHS context; or,
► Sources which contribute to the study aims and are conducted in contexts with similarities to the NHS (eg, universal, publicly-funded healthcare systems).

Minor contribution:
► Sources conducted in non-UK healthcare systems that are markedly different to the NHS (eg, fee-for-service, private insurance scheme systems) but where the mechanisms causing or moderating unprofessional behaviours could plausibly operate in the context of those working in the NHS; or,
► Sources which contribute to the study aims and can clearly help to identify mechanisms which could plausibly operate in the context of the NHS (eg, law, police and army).

We will prioritise the papers of major relevance in relation to the above criteria. These sources will then be sorted into the above categories in a reference manager and assessed for their ability to inform the refinement of programme theories (theoretical relevancy and conceptual richness). If we are unable to develop and refine aspects of the programme theory due to scarcity in majorly relevant sources, we will then draw on literature from the minor relevancy criteria above.

### Step 4: data extraction

In keeping with a realist retroductive analysis, relevant sections of texts that have been interpreted as related to contexts, mechanisms and/or their relationships to outcomes will be coded and organised in NVivo V.12 software (QSR International) . This coding will be both inductive (codes created to categorise data reported in included sources) and deductive (codes created in advance of data extraction and analysis as informed by the initial programme theory). Each new element of relevant data will be used to test and refine aspects of the programme theory, and as it is refined, included sources will be re-scrutinised to search for data relevant to the revised programme theory that may have been missed initially. The characteristics of the included sources will be extracted separately into an Excel spreadsheet to provide a descriptive overview of (but not limited to) settings, intervention, participants and context.

As outlined above, we will start the coding and analysis process using the literature that has been deemed to make a 'major', that is, most relevant contribution to the research questions to start building and refining our programme theory, while progressively focusing the review. Articles categorised as providing 'minor' contributions will be held back and analysed if there is no 'major' study which sheds light on certain aspects of the programme theory. The aim of the review will be to reach theoretical saturation in relation to the objectives, rather than to aggregate every single study that exists in the area. All study-related decisions will be documented and recorded as part of an audit trail to increase transparency and ensure consistency and rigour.

### Step 5: synthesising evidence and drawing conclusions

Our data analysis will use realist logic to make sense of the initial programme theory. Data will be interrogated at individual, team and organisational levels to establish their relationships. This type of analysis will enable us to understand how the most relevant and important mechanisms work in different contexts, allowing us to build more transferable CMOCs—these will be fully formulated by the end of this step. During the review, we will move iteratively between the analysis of particular examples from the literature, refinement of programme theory and further iterative searching for data to test particular subsections of the programme theory as required (figure 1). We will also use the following strategies to make sense of the data (box 1).[50 51]

Many of these processes are analogous to the identification of demi-regularities ('semi-predictable patterns or pathways of programme functioning' across studies). This type of thinking will be essential to help identify underlying mechanisms that are common to different strategies.[40 52] As required, we will also identify and incorporate middle range theories (eg, theories around, for example, groupthink and psychological safety) to enable us to move beyond description and understand the 'set of assumptions' underlying the observed associations.[53]

**Table 2** Dissemination and pathways to impact and who is reached[55]

| Output | Description | Timescale to benefit | How will impact be achieved? | Who is reached? |
|---|---|---|---|---|
| PPI/stakeholder engagement. | PPI and our stakeholder representatives will be actively involved in the production of all outputs. The stakeholder group, including healthcare staff and PPI representatives, will be encouraged to think about alternative or additional approaches to dissemination. | During project and up to 6 months after. | Engaging with the stakeholder group will enable us to understand how best to reach people like them in the most well-targeted manner. | Internal stakeholder group of study and their immediate contacts. |
| Media engagement strategy. | We will identify the most appropriate way to engage with our non-academic stakeholder groups. eg, through engagement with relevant professional bodies (eg, British Medical Association, General Medical Council, Royal College of Nursing, Royal College of Midwives, NHS Employers) and through promoting our findings via alternative publication routes (eg, Health Services Journal, Nursing Times/Standard, BMJ, The Conversation, Twitter). | During project and up to 2 years after. | Engaging with media will allow lighter touch but wider dissemination of our key messages. | Public, academics, practitioners, managers, leaders, policymakers. |
| Resources for NHS managers/leaders. | Tailored resources for NHS managers/leaders, and organisations supporting the healthcare workforce regarding how to implement strategies to reduce unprofessional behaviours. | End of project onwards. | Effects will be visible once stakeholders are able to implement changes and evaluate the impact of those changes. | Managers, leaders. |
| Plain English summaries. | We will create plain English summaries tailored to different audiences. | End of project and up to 2 years thereafter. | This will achieve impact through knowledge transfer in the short-term to medium-term (1 month 2 years). | Healthcare professionals, managers, leaders, training providers, policymakers. |
| Innovative forms of communication. | We have had positive experiences of involving film makers and using the medium of theatre to perform research findings.[56] | 1–5 years. | Such outputs will create a longer-lasting impression with stakeholders than other more traditional forms of dissemination. | Healthcare professionals, managers, leaders, training providers, policymakers. |
| Academic outputs. | In addition to the main project report, an overall findings paper will be submitted to a high-impact peer-reviewed journal, and conference presentations at healthcare staff well-being conferences (such as Health Services Research UK) will be pursued. | 3–5 years. | Informing the agenda for debate and action in health services and in public policy more widely. | Academics, managers, policymakers. |

NHS, National Health Service; PPI, Patient and public involvement.

### Step 6: testing and refining, and developing resources for stakeholders

We will then test and refine our programme theories with our stakeholder group. Informed by the 'Evidence Integration Triangle'[54] and stakeholder involvement, we will use findings from our realist review to produce actionable evidence to support NHS managers/leaders to better understand how work environments may help or hinder unprofessional behaviours and identify what strategies work where.

### ETHICS AND DISSEMINATION
### Dissemination

Dissemination activities comprise step 7 of our project. The project will produce six major types of output in collaboration with our stakeholder group and will present some of these at our end of study project dissemination workshop (table 2).

### Ethical approval

The University of Surrey's Ethics committee have confirmed that ethical review is not needed for this realist review of secondary evidence.

**Contributors** JM led the conceptualisation and development of the overall study design and drafting of the protocol on which this paper is based. JAA drafted the manuscript in dialogue with JM and all authors and helped to refine the communication of the study methodology. RA, MP, JMW, AJ, RM and JW all contributed to the study conceptualisation and design of the study protocol. All

authors substantively revised drafts of this manuscript and approved the final version.

**Funding** This project was supported by the NIHR HS&DR programme with grant number 131606. The views and opinions expressed herein are those of the authors and do not necessarily reflect those of the HS&DR programme.

**Competing interests** None declared.

**Patient and public involvement** Patients and/or the public were involved in the design, or conduct, or reporting, or dissemination plans of this research. Refer to the Methods section for further details.

**Patient consent for publication** Not applicable.

**Ethics approval** Not applicable.

**Provenance and peer review** Not commissioned; externally peer reviewed.

**ORCID iDs**
Jill Maben http://orcid.org/0000-0002-6168-0455
Justin Avery Aunger http://orcid.org/0000-0001-6975-4570
Mark Pearson http://orcid.org/0000-0001-7628-7421
Judy M Wright http://orcid.org/0000-0002-5239-0173
Aled Jones http://orcid.org/0000-0002-2921-8236

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
