## [Reviewer comments · BMJ Open]

ARTICLE DETAILS

TITLE (PROVISIONAL)	Why do acute healthcare staff engage in unprofessional behaviours towards each other and how can these behaviours be reduced? A realist review protocol
AUTHORS	Maben, Jill; Aunger, Justin; Abrams, Ruth; Pearson, Mark; Wright, Judy; Westbrook, Johanna; Mannion, Russell; Jones, Aled

VERSION 1 – REVIEW

REVIEWER	Thomas Gray Sheffield Teaching Hospitals NHS Foundation Trust
REVIEW RETURNED	09-Mar-2022

GENERAL COMMENTS	This study protocol provides a clear context for the research question and is well referenced with appropriate and up to date sources. The methodology is very clear with the process laid out, the systematic review has been registered with PROSPERO and the methods are replicable. My only comment is that one of the headings reads: 'Prevalence of unprofessional behaviours in the UK healthcare'. I think the word system is missing. I did not identify any other typographical errors. I look forward to reading the completed study.
---

REVIEWER	Linda Guo Mater Health Services Brisbane
REVIEW RETURNED	13-Mar-2022

GENERAL COMMENTS	Overall, I see the value in developing a mechanistic understanding of these behaviours and mitigation strategies and look forward to reading the findings of this work; however, there are elements of the methodology that require some additional clarification. General comments: Please include further details of the search strategy (databases and other sources to be used, inclusion/exclusion criteria) and screening process in the Methods and Analysis section of the abstract. Regarding the phrase 'how and why [mitigation] strategies work' used in multiple parts of the manuscript: Do the questions of 'how' and 'why' an intervention works have substantively different answers? If not, only 'how' is required. Please include the dates of the study in the manuscript as per journal guidelines. The Discussion and Conclusion is very similar in content to the
--

	introduction – consider moving any unique information here to the introduction and removing the section. Specific comments: Page 2, line 55: I feel this sentence could be simplified to something such as, ‘We seek to build upon these and gain further understanding of the range, causes and effects of these behaviours, as well as how mitigation strategies may work, in which contexts and for whom.’ Page 3, line 43: ‘This review is the first...’, Thankfully, the use of broader terminology than bullying and harassment is now occurring more commonly (e.g. The Effect of Health Care Professional Disruptive Behavior on Patient Care: A Systematic Review by Hicks et al). The start of this sentence should therefore be updated, perhaps to ‘This review is the first to identify the causes of unprofessional behaviours...’. Also please clarify what is meant by contexts here and the expected range of these will be examined? Is the context distinct from the study setting? If they are the same, then the language should be consistent across the manuscript. Page 3, line 46: The second and third dot point can be combined as they both express that developing an understanding of underlying mechanisms is an advantage of the realist review methodology. Page 3, line 55: As the anticipated strengths of the realist methodology have been explored, I would imagine that there would equally be some additional methodological limitations that could be added here, such as publication bias (due to reliance on existing published evidence) and selection bias of studies that align with the theories Introduction: The first two paragraphs of the introduction can be simplified to focus on the definition/what is meant by ‘unprofessional behaviour’, as the remaining sentences introduce themes that are discussed in further detail later on (impacts, prevalence, aims of the review). Page 3, line 18: Remove ‘aren’t’ (colloquial) by changing to ‘unprofessional behaviours are being increasingly recognised as unacceptable’ Page 3, line 19: Remove ‘especially’ Page 3, line 23: Remove ‘for example, misogynist and racist harassment’, as this is purely a repetition of the groups that have been introduced at the beginning of the sentence (women and members of minority communities). Page 5, line 58: Simplify to ‘incivility within healthcare teams, including high workload’ Page 6, line 5: Change to ‘concluded that the involvement of management and wider organisational commitment were key to intervention success’. Page 6, line 16: Change to ‘detail regarding the causes of such behaviours’ Page 6, line 17: The phrasing of the aim of the review could be more
--	--

	direct. Could it be simplified to something like 'how, why and in what circumstances these behaviours occur, and how they can be mitigated, managed and prevented'? Page 6, line 19: These deficits in the existing literature should be either moved to the previous paragraph with the others, or rephrased as an aim for your review (the topic of this paragraph). Methods: Page 6, line 36: As above, simply to something such as 'how, why and in what circumstances these behaviours occur, and how they can be mitigated, managed and prevented'. Page 7, line 11: Change to 'How are unprofessional behaviours defined...' Page 7, line 13: Change to 'How do current strategies...' Page 7, line 54: 'between and within' – I am not sure if the words 'and within' add any further information Page 11, line 6: Change to 'more informal, purposive searches, citation tracking and reference scanning.' Page 11, line 24: The second half of this sentence ('to shed further light' onwards) can be deleted without much loss of meaning, as it is not specific to the search strategy and mirrors the objectives of the study, which are well covered above. Page 11, line 35: These considerations are effectively exclusion criteria – they should be moved to the 'Screening and Inclusion Criteria' paragraph after the inclusion criteria, and included in Table 1. Page 12, line 39: Screening of search results will be undertaken by JA in collaboration with JM and RA.' Please clarify if the screening of search results will be done independently by two reviewers? Page 13, line 5: Change to 'Decisions regarding the inclusion and exclusion of sources will be based' Page 13, line 9: Please specify if assessment of rigour will include an assessment of methodological quality/robustness beyond the detail in which the methods are described. Page 13, line 6: It appears that the studies are first screened against the inclusion criteria, and then secondarily screened for rigour and relevance (based on contribution to the programme theories) – is that correct? If this is the case, please remove 'based on inclusion criteria' for clarity. Page 15, line 4: Change 'lying behind' to 'underlying' or 'underpinning' Page 15, line 9: Remove 'emerging evidence informed'
--	---

VERSION 1 – AUTHOR RESPONSE

Reviewer 1	
This study protocol provides a clear context for the research question and is well referenced with appropriate and up to date sources. The methodology is very clear with the process laid out, the systematic review has been registered with PROSPERO and the methods are replicable. My only comment is that one of the headings reads: 'Prevalence of unprofessional behaviours in the UK healthcare'. I think the word system is missing. I did not identify any other typographical errors. I look forward to reading the completed study.	Thank you for your positive feedback regarding our study protocol and we appreciate your feedback regarding the heading title which we have now corrected.
Reviewer 2	
Overall, I see the value in developing a mechanistic understanding of these behaviours and mitigation strategies and look forward to reading the findings of this work; however, there are elements of the methodology that require some additional clarification.	Thank you for the kind words regarding the study and we very much appreciate the time you have spent on helping to improve the manuscript.
Please include further details of the search strategy (databases and other sources to be used, inclusion/exclusion criteria) and screening process in the Methods and Analysis section of the abstract.	Unfortunately, we simply do not have sufficient word count to include a great deal of detail about the inclusion/exclusion criteria in the abstract without sacrificing significantly elsewhere, however, we have optimised the wording to include further information on databases we are searching.
Regarding the phrase 'how and why [mitigation] strategies work' used in multiple parts of the manuscript: Do the questions of 'how' and 'why' an intervention works have substantively different answers? If not, only 'how' is required.	Whilst we understand why this might be unclear, we do think it important to keep both 'how' and 'why'. In a realist project the 'how' is often used in terms of understanding the components of the programme (the mechanisms by which the programme 'works') that are intended to produce the desired effects. In contrast, the 'why' helps to explain why, in different circumstances, the same programme might not deliver the same results – due to contextual factors influencing its implementation.
Please include the dates of the study in the manuscript as per journal guidelines.	The study commenced October 2021 for 18 months and is expected to finish in March 2023. However, we have not included the study dates in the manuscript at present because it was a review protocol but would be happy to do so if the editor requires.

The Discussion and Conclusion is very similar in content to the introduction – consider moving any unique information here to the introduction and removing the section.	We have removed this section as suggested and since the editor notes that a Conclusions section is not a requirement for study protocols..
Page 2, line 55: I feel this sentence could be simplified to something such as, ‘We seek to build upon these and gain further understanding of the range, causes and effects of these behaviours, as well as how mitigation strategies may work, in which contexts and for whom.’	We have now simplified this sentence somewhat to remove any redundant language.
Page 3, line 43: ‘This review is the first...’, Thankfully, the use of broader terminology than bullying and harassment is now occurring more commonly (e.g. The Effect of Health Care Professional Disruptive Behavior on Patient Care: A Systematic Review by Hicks et al). The start of this sentence should therefore be updated, perhaps to ‘This review is the first to identify the causes of unprofessional behaviours...’. Also please clarify what is meant by contexts here and the expected range of these will be examined? Is the context distinct from the study setting? If they are the same, then the language should be consistent across the manuscript.	We have made changes to the sentence in line with suggested edits. The contexts for the causes refer more to the settings in which these unprofessional behaviours emerge, whereas in the case of strategies the contexts are indeed referring to the circumstances in which the programmes are implemented.
Page 3, line 46: The second and third dot point can be combined as they both express that developing an understanding of underlying mechanisms is an advantage of the realist review methodology.	We agree and have now created a new point to replace this which highlights the benefits and uniqueness of reviewing both causes and strategies in one study (pg 2).
Page 3, line 55: As the anticipated strengths of the realist methodology have been explored, I would imagine that there would equally be some additional methodological limitations that could be added here, such as publication bias (due to reliance on existing published evidence) and selection bias of studies that align with the theories	Compared to most approaches to systematic reviewing, a realist review draws on a very broad range of sources including grey literature - so arguably doesn't suffer from publication bias (aka, not including evidence from non-peer-reviewed sources). We are also engaging with our stakeholder group throughout the review, part of which is to draw on their knowledge to identify other important sources, whether grey literature or 'other'. In terms of selection bias, we are not seeking to include literature (at least at the systematic stage) that will only ratify our initial theories. Any purposive searching that we would perform would be conducted in a manner that could refute or support our existing theories. In the 'Summary' section we state three limitations which we feel sufficiently represent the limitations of the review.
Introduction: The first two paragraphs of the introduction can be simplified to focus on the definition/what is meant by ‘unprofessional behaviour’, as the	We agree and have consolidated the two paragraphs into one.

remaining sentences introduce themes that are discussed in further detail later on (impacts, prevalence, aims of the review).	
Page 3, line 18: Remove 'aren't' (colloquial) by changing to 'unprofessional behaviours are being increasingly recognised as unacceptable' Page 3, line 19: Remove 'especially' Page 3, line 23: Remove 'for example, misogynist and racist harassment', as this is purely a repetition of the groups that have been introduced at the beginning of the sentence (women and members of minority communities). Page 5, line 58: Simplify to 'incivility within healthcare teams, including high workload' Page 6, line 5: Change to 'concluded that the involvement of management and wider organisational commitment were key to intervention success'. Page 6, line 16: Change to 'detail regarding the causes of such behaviours'	These line-specific edits have been performed.
Page 6, line 17: The phrasing of the aim of the review could be more direct. Could it be simplified to something like 'how, why and in what circumstances these behaviours occur, and how they can be mitigated, managed and prevented'? Page 6, line 19: These deficits in the existing literature should be either moved to the previous paragraph with the others, or rephrased as an aim for your review (the topic of this paragraph).	We have rephrased the aim to 'This realist review aims to improve context-specific understanding of how, why and in what circumstances unprofessional behaviours between staff in acute healthcare settings occur, and how strategies can be implemented to mitigate, manage, and prevent them'. We have now adjusted Page 16 line 19 as suggested and removed the repetitive aspects of this second paragraph.
Methods: Page 6, line 36: As above, simply to something such as 'how, why and in what circumstances these behaviours occur, and how they can be mitigated, managed and prevented'. Page 7, line 11: Change to 'How are unprofessional behaviours defined...' Page 7, line 13: Change to 'How do current strategies...'	We have made these line-specific edits as pointed out by the reviewer.

Page 7, line 54: 'between and within' – I am not sure if the words 'and within' add any further information Page 11, line 6: Change to 'more informal, purposive searches, citation tracking and reference scanning.' Page 11, line 24: The second half of this sentence ('to shed further light' onwards) can be deleted without much loss of meaning, as it is not specific to the search strategy and mirrors the objectives are the study, which are well covered above.	
Page 11, line 35: These considerations are effectively exclusion criteria – they should be moved to the 'Screening and Inclusion Criteria' paragraph after the inclusion criteria, and included in Table 1. Page 12, line 39: Screening of search results will be undertaken by JA in collaboration with JM and RA.' Please clarify if the screening of search results will be down independently by two reviewers? Page 13, line 5: Change to 'Decisions regarding the inclusion and exclusion of sources will be based'	We have moved the exclusion criteria text to the more correct location immediately below table 1. We have also clarified that JA will perform the majority of screening with RA, JM, or MP performing a screening of 10% of the subsample to ensure accuracy (pg. 10). We have corrected the relevancy part (end of pg 10/beginning of pg 11) to refer to the major/minor criteria as opposed to the inclusion criteria. To further clarify our method we have moved all discussion of the selection process to the 'Step 3: Article selection' section (which also includes Table 1: Inclusion criteria) which we believe makes more sense.
Page 13, line 9: Please specify if assessment of rigour will include an assessment of methodological quality/robustness beyond the detail in which the methods are described. Page 13, line 6: It appears that the studies are first screened against the inclusion criteria, and then secondarily screened for rigour and relevance (based on contribution to the programme theories) – is that correct? If this is the case, please remove 'based on inclusion criteria' for clarity. Page 15, line 4: Change 'lying behind' to 'underlying' or 'underpinning'	Assessment of rigour will be limited to what is described in this section, which is typical of realist reviews, where evidence is generally included if it is useful to generating theory – as long as it is considered trustworthy. We have made substantial edits to the rigour section for clarity (pg 11). Your assertion that studies are screened against inclusion criteria and later rigour and relevance was correct. As such, we have made edits throughout the screening section, which now has its own heading, to enhance clarity.

Page 15, line 9: Remove 'emerging evidence informed'	We have also made these two line-specific edits. Thank you for all of your suggestions and edits and hope you agree that we have implemented them appropriately.
--	--

VERSION 2 – REVIEW

REVIEWER	Linda Guo Mater Health Services Brisbane
REVIEW RETURNED	02-May-2022

GENERAL COMMENTS	I congratulate the authors on their attention to detail and the re-submission of a much improved manuscript. In particular, restructuring the methodology section has made a substantial difference to the readability of the paper. The suggestions below are minor and largely grammatical: Page 1, line 37: Mark Pearson's name is typed twice under 'Email addresses' Page 3, line 19: The last sentence of the first paragraph can be removed as it repeats two concepts that have already been covered: prevalence in the sentence two of the paragraph and variable intensity of the behaviours in sentences four and five. Page 5, line 7: '[...] to produce knowledge to enable more detail' – should this be 'to produce more detailed knowledge'? Page 5, line 11: Change to 'in what circumstances unprofessional behaviours between staff in acute healthcare settings occur, and how they can be best understood' Page 6, line 8: Change semicolon after well-being to a comma for consistency Page 12, line 27: as it has been previously introduced, the CMOC abbreviation should be used
---

VERSION 2 – AUTHOR RESPONSE

Reviewer Comment	Author Response
Reviewer 2	
Page 1, line 37: Mark Pearson's name is typed twice under 'Email addresses' Page 3, line 19: The last sentence of the first paragraph can be removed as it repeats two concepts that have already been covered: prevalence in the sentence two of the paragraph and variable intensity of the behaviours in sentences four and five.	We have corrected these grammatical issues.

Page 5, line 7: '[...] to produce knowledge to enable more detail' – should this be 'to produce more detailed knowledge'? Page 5, line 11: Change to 'in what circumstances unprofessional behaviours between staff in acute healthcare settings occur, and how they can be best understood'	We have made these edits as suggested.
Page 6, line 8: Change semicolon after well-being to a comma for consistency Page 12, line 27: as it has been previously introduced, the CMOC abbreviation should be used	We have corrected these errors.